# Modulation of Ceramide-Induced Apoptosis in Enteric Neurons by Aryl Hydrocarbon Receptor Signaling: Unveiling a New Pathway beyond ER Stress

**DOI:** 10.3390/ijms25168581

**Published:** 2024-08-06

**Authors:** Mallappa Anitha, Supriya M. Kumar, Imhoi Koo, Gary H. Perdew, Shanthi Srinivasan, Andrew D. Patterson

**Affiliations:** 1Department of Veterinary and Biomedical Sciences, The Pennsylvania State University, University Park, PA 16802, USA; azv2@psu.edu (M.A.); iuk41@psu.edu (I.K.); ghp2@psu.edu (G.H.P.); 2Department of Digestive Diseases, Emory University School of Medicine, Atlanta, GA 30322, USA; ssrini2@emory.edu; 3Atlanta VA Medical Center, Decatur, GA 30033, USA

**Keywords:** TCDD, AHR, ENS, ceramides, cytotoxicity, apoptosis

## Abstract

2,3,7,8-tetrachlorodibenzo-*p*-dioxin (TCDD), a persistent organic pollutant and a potent aryl hydrocarbon receptor (AHR) ligand, causes delayed intestinal motility and affects the survival of enteric neurons. In this study, we investigated the specific signaling pathways and molecular targets involved in TCDD-induced enteric neurotoxicity. Immortalized fetal enteric neuronal (IM-FEN) cells treated with 10 nM TCDD exhibited cytotoxicity and caspase 3/7 activation, indicating apoptosis. Increased cleaved caspase-3 expression with TCDD treatment, as assessed by immunostaining in enteric neuronal cells isolated from WT mice but not in neural crest cell-specific *Ahr* deletion mutant mice (*Wnt1Cre^+/−^/Ahr^b(fl/fl^*^)^), emphasized the pivotal role of AHR in this process. Importantly, the apoptosis in IM-FEN cells treated with TCDD was mediated through a ceramide-dependent pathway, independent of endoplasmic reticulum stress, as evidenced by increased ceramide synthesis and the reversal of cytotoxic effects with myriocin, a potent inhibitor of ceramide biosynthesis. We identified *Sptlc2* and *Smpd2* as potential gene targets of AHR in ceramide regulation by a chromatin immunoprecipitation (ChIP) assay in IM-FEN cells. Additionally, TCDD downregulated phosphorylated Akt and phosphorylated Ser9-GSK-3β levels, implicating the PI3 kinase/AKT pathway in TCDD-induced neurotoxicity. Overall, this study provides important insights into the mechanisms underlying TCDD-induced enteric neurotoxicity and identifies potential targets for the development of therapeutic interventions.

## 1. Introduction

The interplay between environmental pollutants and biological systems has garnered considerable attention due to the pervasive nature and often long-lasting effects of these compounds. Among them, 2,3,7,8-tetrachlorodibenzo-*p*-dioxin (TCDD), a member of the dioxin family, is a persistent organic pollutant [1] known for its detrimental effects on human health and the environment due to its high lipophilicity and resistance to metabolic clearance [2]. The adverse effects of TCDD span carcinogenesis, immunotoxicity, reproductive and developmental abnormalities, and damage to the nervous system [3,4,5,6,7,8], underscoring the need for an in-depth understanding of its mechanisms of action. Of particular concern is its implications for delayed intestinal motility and enteric neuron survival, pivotal for gastrointestinal functionality [9,10].

The aryl hydrocarbon receptor (AHR), a ligand-activated transcription factor that belongs to the bHLH-PAS family [11], is a mediator of TCDD toxicity, influencing apoptosis, cell cycle regulation, and immunomodulation [12,13,14,15]. While the role of AHR in environmental toxicology and carcinogenesis is well documented [16,17], emerging evidence suggests its involvement in apoptosis regulation [18,19], offering new perspectives on tissue homeostasis and mitigation of pathological conditions. This is of importance considering that dysregulation of apoptosis can contribute to various pathological conditions, including cancer, neurodegenerative diseases, and autoimmune disorders [20,21].

Endoplasmic reticulum (ER) stress and apoptosis have been implicated in various pathological conditions in different cell types, including neurons [22,23,24,25]. Studies have shown that AHR can induce apoptosis through ER stress in neurons [26]. Ceramides, key sphingolipid signaling molecules, are involved in various cellular processes, including cell growth, differentiation, and apoptosis [27,28,29]. There are three main pathways for ceramide production [30]: The de novo synthesis of ceramide occurs in the endoplasmic recticulum (ER) [31,32] and in mitochondria [33,34] with the condensation of palmitoyl-CoA and serine to form 3-ketosphinganine by serine palmitoyltransferase (SPT), and then produces ceramide by the action of ceramide synthase (CerS) and dihydroceramide desaturases (Des). The sphingomyelinase (SMase) pathway in the plasma membrane produces ceramides by the hydrolysis of sphingomyelin by the enzyme neutral sphingomyelinase (N-SMase) [35]. Ceramides can be reformed by the salvage pathway [36] that involves many enzymes, such as acid sphingomyelin phosphodiesterase (A-SMase), acid ceramidases (A-CDase), and glycosyl synthase (GCS) within the endo-lysosomal route [30]. Accumulation of ceramides has been observed in cells treated with various apoptotic agents, such as TNF-alpha, ionizing radiation, UV light, and chemotherapeutic agents, suggesting a ceramide-mediated role in cellular responses to these agents [37,38,39,40]. Exogenous addition of cell-permeable ceramide analogs induces apoptosis in a variety of cell lines [41,42,43]. Ceramide can modulate cell death by inhibiting the PI3-Kinase antiapoptotic/prosurvival pathway, which involves the regulation of Akt activity [44]. Majumder et al. have shown by genome-wide CRISPR/Cas9 screens that AHR stimulates sphingolipid levels [45]. Recent studies have demonstrated that persistent organic pollutants like TCDD, polychlorinated biphenyls (PCBs), and tetrachlorodibenzofuran (TCDF) regulate sphingolipid metabolism [46,47].

The specific role of AHR in ceramide-induced apoptosis is not well understood. Our recent studies have shown that AHR activation by TCDD affects enteric neuronal survival and delays intestinal motility in mice [9]. In this current study, we sought to extend our research by looking at the underlying molecular mechanisms of TCDD-mediated enteric neurotoxicity, the signaling pathways involved, and its molecular targets using IM-FEN cells [48]. We utilized different doses of TCDD (0.1, 1, and 10 nM) at various time points (3, 6, 12, and 24 h) to assess its effects on cell survival, ER stress markers, ceramide biosynthesis-related genes, N-SMase activity, and ceramide levels. Human exposure to TCDD typically occurs at low concentrations through contaminated food, water, and air. The levels of TCDD found in the general population are usually in the picomolar (pM) to low nanomolar (nM) range. According to the Centers for Disease Control and Prevention (CDC), background levels of TCDD in human serum typically range from 3 to 7 parts per trillion (ppt). A study by Pavuk et al. found a mean serum TCDD concentration of 5.8 ppt (~0.02 nM) in a background group of Vietnam veterans who were exposed to Agent Orange, while the mean TCDD in the low and high categories were substantially higher, 15.6 and 69.4 ppt, respectively, which corresponds to 0.05 nM and 0.2 nM [49]. Our approach helps to elucidate the potential neurotoxic effects of TCDD across different exposure levels, providing valuable insights into the risks posed by this environmental pollutant and the underlying mechanisms of its toxicity. We also examined the involvement of AHR in TCDD-induced cytotoxicity by using CH-223191, a potent and specific AHR antagonist, and primary enteric neuronal cultures from WT and neural crest cell-specific *Ahr^−/−^* mice (*Wnt1Cre^+/−^/Ahr^b(fl/fl)^*). Further, we investigated the PI3K/Akt signaling pathway, which plays an important role in the regulation of enteric neuronal cell survival. Our findings suggest that AHR contributes to apoptosis in IM-FEN cells by increasing ceramide levels, which subsequently inhibits the PI3K antiapoptotic/prosurvival pathway.

## 2. Results

### 2.1. TCDD-Induced Apoptosis in IM-FEN Cells

Our laboratory has previously shown that TCDD affects IM-FEN cell survival but not proliferation [9]. This data was further supported by Western blotting showing increased expression of cleaved caspase-3, an important mediator of the apoptosis process, and cleaved PARP, which is involved in DNA repair and whose cleavage by caspases is a hallmark of apoptosis, following TCDD treatment at concentrations of 0.1, 1, and 10 nM for 24 h, indicating significant induction of apoptosis (*p* < 0.0001; Figure 1A,B). Immunofluorescence staining for TUJ1, a neuronal marker, in conjunction with TUNEL staining, which detects apoptotic cells by labeling DNA strand breaks with fluorescently labeled nucleotides, further validated these findings. IM-FEN cells treated with 10 nM TCDD for 24 h exhibited more apoptotic cells with condensed (fragmented) nuclei compared to vehicle treated cells (Figure 1C). This data underscores the critical role of TCDD in promoting neuronal apoptosis, offering insights into the mechanisms underlying TCDD-induced neurotoxicity.

### 2.2. AHR-Dependent TCDD-Induced Cytotoxicity and Apoptosis in IM-FEN Cells

The AHR antagonist CH-223191 preferentially blocks TCDD binding to the AHR and subsequent heterodimerization with ARNT and thus AHR-dependent mediated target gene expression [50]. To explore the involvement of AHR in TCDD-elicited cytotoxicity and apoptosis, cells were pretreated with 10 µM CH-223191 for 1 h and then treated with Veh or 10 nM TCDD at different time points (3, 6, 12, and 24 h). Cytotoxicity was assessed by LDH release in the culture medium using the LDH-Glo Cytotoxicity Assay kit, and apoptosis was examined by assessing caspase-3/7 activity using the Caspase-Glo 3/7 Assay kit. TCDD induced a gradual increase in LDH release in the culture medium of IM-FEN cells compared to vehicle-treated cells. CH-223191 significantly mitigated the cytotoxic effects of TCDD, aligning LDH levels with those observed in vehicle-treated cells (Figure 2A), indicating that AHR was involved in the TCDD-induced effect. Additionally, caspase 3/7 activity, a marker of apoptosis, exhibited a notable increase 6 h post-TCDD exposure, diminishing at subsequent time points (12 and 24 h). CH-223191 pre-treatment effectively counteracted the rise in caspase 3/7 activity triggered by TCDD, further emphasizing the dependency of TCDD’s apoptotic effects on AHR signaling (Figure 2B).

Further, primary culture cells (myenteric neurons) from *Wnt1Cre^+/−^/Ahr^b(fl/fl)^* and *Wnt1Cre^−/−^/Ahr^b(fl/fl)^* mice were treated with vehicle and 10 nM TCDD for 24 h at 37 °C. Subsequently, apoptosis was examined by staining the cells with TUJ1 and cleaved caspase-3, which served as markers for neuronal cells and apoptosis, respectively. Cleaved caspase-3 expression was elevated in 10 nM TCDD-treated myenteric cells isolated from *Wnt1Cre^−/−^/Ahr^b(fl/fl)^* control mice compared to vehicle-treated cells (Figure 2C). In contrast, myenteric neurons from *Ahr*^−/−^ mice, *Wnt1Cre^+/−^/Ahr^b(fl/fl)^* mice displayed no significant change in cleaved caspase-3 expression following TCDD treatment, highlighting the indispensable role of AHR in facilitating TCDD-induced neuronal apoptosis (Figure 2D).

### 2.3. TCDD-Induced Apoptosis in IM-FEN Cells Proceeds via an ER Stress-Independent Pathway

IM-FEN cells were treated with vehicle or 10 nM TCDD for 24 h. Additionally, some cells were pre-treated with the AHR antagonist CH-223191 (10 µM) for one hour prior to a 24 h exposure to 10 nM TCDD. To benchmark ER stress, thapsigargin (300 nM, 5 h treatment) was used as a positive control. To investigate the potential role of ER stress in apoptosis induction, the expression of ER stress-associated proteins such as 78-kDa glucose-regulated protein (GRP78), inositol-requiring enzyme 1α (IRE1α), and the transcription factor C/EBP-homologous protein (CHOP) (Figure 3) was examined by Western blotting. The expression levels of these markers did not significantly vary across the treatment groups, corroborating the hypothesis that ER stress pathways are not implicated in the TCDD-induced apoptotic process within IM-FEN cells.

### 2.4. TCDD Regulates Ceramide Metabolism: Insights from Gene Expression, Enzymatic Activity, and Lipidomic Analyses

Following the exclusion of the ER stress mechanism in inducing apoptosis in IM-FEN cells, we examined the impact of AHR on the transcriptional regulation of genes encoding key enzymes involved in the ceramide synthesis pathway by qPCR using primers listed in Table 1 We observed that the expression of ceramide synthesis-related genes, including serine palmitoyltransferse long-chain base subunits (*Sptlc1*, *Sptlc2*), ceramide synthases (*Cers2*, *Cers5*, *Cers6*), dihydroceramide desaturase 1 (*Degs1*), and sphingomyelin phosphodiesterase (*Smpd1*, *Smpd2*, *Smpd3*, *Smpd4*), was significantly increased within 1 to 6 h following treatment with 10 nM TCDD. However, their expression levels decreased at 12 h and 24 h post-treatment (Figure 4A). In addition, we measured the activity of neutral sphingomyelinase (N-SMase), an enzyme involved in ceramide metabolism. We found that N-SMase activity was significantly elevated (Figure 4B, *p* < 0.001) in the IM-FEN cells following 10 nM TCDD treatment.

To delve deeper into the effects of TCDD on the ceramide synthesis pathway, the sphingolipid metabolism of IM-FEN cells treated with vehicle and 10 nM TCDD for 3 h was analyzed via lipidomics. This analysis, represented through heatmaps, uncovered significant differences in the levels of various sphingolipids, including ceramide, sphingomyelin, and hexosylceramide, in IM-FEN cells treated with TCDD compared to vehicle-treated cells (Figure 4C,D). Total ion chromatogram (TIC)-based normalization values were applied to the heatmap.

To further investigate the potential ceramide gene targets of AHR, genomic DNA fragments bound by AHR were analyzed by a Chromatin Immunoprecipitation (ChIP) assay. For the ChIP assay, IM-FEN cells were treated with vehicle and 10 nM TCDD for 1 h. Immunoprecipitated DNA was amplified and quantified by qPCR with primers for *Cyp1a1* (positive control), *Sptlc2*, and *Smpd2* gene promoters containing known and suspected DREs (Table 2). The ChIP-qPCR showed a significant increase in the percentage of *Cyp1a1* in AHR-recognized genomic DNA fragments (Figure 4E, *p* < 0.001) in the TCDD treated cells. The percentage of *Sptlc2* and *Smpd2* in AHR-recognized genomic DNA fragments increased significantly after TCDD treatment (Figure 4F). These results suggest that AHR directly binds to the promoters of *Sptlc2* and *Smpd2*, indicating their regulatory involvement in ceramide synthesis pathway modulation. This direct targeting by AHR underscores the intricate regulation of ceramide metabolism following TCDD exposure.

The results collectively demonstrate that TCDD-induced activation of AHR leads to the upregulation of ceramide synthesis-related genes and an increase in N-SMase activity. These findings highlight a potential mechanistic link between AHR activation and ceramide metabolism in IM-FEN cells, suggesting a pivotal role of AHR in modulating ceramide metabolism.

### 2.5. Induction of Cytotoxicity and Apoptosis by Short-Chain Ceramides in IM-FEN Cells

Ceramides have been found to induce cytotoxicity by triggering apoptosis [44,51]. In our study, we investigated the impact of TCDD treatment on ceramide synthesis and its potential role in apoptosis in IM-FEN cells. We specifically examined the effects of short-chain ceramides (C_2_-ceramide and C_6_-ceramide) on cell viability and apoptosis, as these forms of ceramides can easily penetrate the cell membrane compared to natural long-chain ceramides [42]. To assess cytotoxicity and apoptosis induction, we utilized both cytotoxicity assays measuring lactate dehydrogenase (LDH) release and caspase-3/7 activity assays. IM-FEN cells were exposed to various treatments: vehicle, 25 µM C_2_- and C_6_-ceramides, as well as inactive analogs called C_2_- and C_6_- dihydroceramides (DHC) (25 µM), which served as negative controls. We monitored the cells at different time points: 30 min, and 1, 3, 6, 12, and 24 h. The LDH release assay indicated an increase in cytotoxicity following treatment with C_2_-ceramide and C_6_-ceramide, marked by elevated LDH levels in the medium, suggesting compromised cell membrane integrity. This effect was most pronounced within the first 6 h post-treatment and showed a decline over time, highlighting the temporal dynamics of ceramide-induced cytotoxicity (Figure 5A).

Furthermore, we examined caspase 3/7 activity, which serves as an indicator of apoptosis. With C_2_-ceramide treatment, we observed higher caspase 3/7 activity at 3 and 6 h, while C_6_-ceramide treatment resulted in increased caspase 3/7 activity at 6 and 12 h (Figure 5B). These findings suggest that the timing and magnitude of caspase activation may vary depending on the specific ceramide treatment. In contrast, treatments with C_2_- and C_6_-dihydroceramides (DHC), inactive analogs of ceramides, did not elicit similar cytotoxic or apoptotic responses, underscoring the specificity of ceramide action. The percentage of cytotoxicity and caspase activity for treatments with C_2_- and C_6_-ceramide reaching zero starting at 12 h indicates that maximum cytotoxicity was likely achieved, leading to the death of all cells. This suggests that any remaining cells are either resistant or that the assay reached its maximal detection limit, thus showing no further increase in measured activity. The zero value indicates that additional ceramide treatment does not further impact cell viability or caspase activity, likely due to the saturation of the ceramide effect within the experimental conditions.

### 2.6. TCDD-Induced Cytotoxicity in IM-FEN Cells: The Mediating Role of Ceramides

We found that exposure to TCDD not only precipitates apoptosis in IM-FEN cells but also stimulates the production of ceramides. This linkage was further substantiated by our observations that applying exogenous ceramide directly induces both cytotoxicity and apoptosis in these cells. To delineate the specific role of ceramides in TCDD-induced cytotoxicity in IM-FEN cells, we designed experiments using 10 nM TCDD, applied either in isolation or combined with 10 µM myriocin. Myriocin serves as an inhibitor of serine palmitoyltransferase (SPT), the enzyme that catalyzes the rate-limiting step in the de novo ceramide synthesis pathway. These treatments spanned various durations, namely 3, 6, 12, and 24 h, allowing us to assess the cytotoxic effect through the quantification of lactate dehydrogenase (LDH) release into the culture medium.

Our results clearly demonstrated that TCDD exposure incrementally elevates LDH release in IM-FEN cells when compared to vehicle-treated controls. Interestingly, the co-administration of myriocin with TCDD significantly attenuated the TCDD-induced cytotoxic effects, although not completely negating them (Figure 6). This observation suggests the possible involvement of an alternative ceramide synthesis route, such as the N-SMase pathway, in contributing to the cytotoxic effects seen in IM-FEN cells under TCDD influence. These findings highlight the pivotal role of ceramides in mediating the adverse effects of TCDD on the viability of IM-FEN cells, emphasizing the complexity of the underlying mechanisms.

### 2.7. TCDD and Ceramide Impair Neuronal Survival Pathway by Modulating Akt and GSK-3β Signaling

We have previously shown that exposure to TCDD results in the loss of enteric neurons [9]. This detrimental effect is thought to be mediated by diminished AKT-mediated survival signaling, thereby activating GSK-3β, two critical components of neuronal cell survival pathways. Additionally, reports have pointed to the influence of ceramide on the PI3K/AKT pathway, with ceramide-induced neuronal apoptosis linked to the dephosphorylation of AKT and GSK-3β [52,53].

Our study sought to delve into the underlying molecular mechanism responsible for neuronal cell death caused by TCDD and ceramide, with a specific focus on the PI3K/AKT/GSK-3β pathway. To investigate this further, IM-FEN cells were exposed to varying concentrations of TCDD (0.1, 1, and 10 nM) for 24 h, and the levels of phosphorylated AKT and GSK-3β were measured using Western blotting.

The results showed a significant decrease in AKT phosphorylation, which is directly correlated with an increase in the active form of GSK-3β due to its dephosphorylation in the TCDD-treated IM-FEN cells (*p* < 0.0001; Figure 7A,B). This correlation underscores the capability of TCDD to interfere with the AKT-mediated survival signaling pathway, thereby potentiating the role of GSK-3β in driving neuronal apoptosis.

In a parallel study, the impact of ceramide on the AKT survival pathway was examined by treating IM-FEN cells with a vehicle and 25 μM C_2_ ceramide for 24 h. Analysis of AKT and GSK-3β phosphorylation levels by Western blotting revealed that C_2_ ceramide significantly decreased their phosphorylation (*p* < 0.0001; Figure 7C,D). This indicates that ceramide affects the AKT survival pathway by activating GSK-3β, which is known to induce apoptosis by activating a cascade of reactions involving caspases. These results support the notion that apoptosis in IM-FEN cells induced by TCDD may operate through the suppression of the PI3K/AKT/GSK-3β survival pathway, with a potential regulatory role played by ceramides.

## 3. Discussion

We have previously reported that AHR activation by TCDD in the enteric nervous system (ENS) adversely affects nitrergic neurons, leading to delayed intestinal motility due to TCDD-induced cytotoxicity [9]. In this study, we unveil new insight into the mechanism by which TCDD, a persistent environmental pollutant, induces neurotoxicity via AHR. Our findings demonstrate that TCDD exposure leads to significant apoptosis in IM-FEN cells through a ceramide accumulation pathway, independent of the traditionally implicated endoplasmic reticulum (ER) stress mechanisms. We identified that AHR directly regulates the expression of *Sptlc2* and Smpd2 in IM-FEN cells, key genes involved in ceramide biosynthesis, highlighting a critical role for AHR in modulating lipid metabolism pathways beyond its established functions. Furthermore, our study elucidates the impact of TCDD on the PI3K/Akt signaling pathway, showing that TCDD exposure results in decreased phosphorylation of AKT and GSK-3β, key players in cell survival signaling. Taken together, our findings improve understanding of the toxicological mechanisms of TCDD, revealing a novel AHR-mediated pathway that leverages ceramide accumulation to induce apoptosis in enteric neurons, independent of ER stress, and underscores the complexity of the role of AHR in cellular and environmental health.

The AHR controls not only the cellular response to a number of toxic and carcinogenic compounds but also a myriad of physiological functions [54]. Studies have reported that TCDD-induced activation of apoptotic signals occurs in NGF-differentiated pheochromocytoma (dPC12) cells, mouse cerebellar granule cells, primary cultures of cerebral cortical neurons, and zebrafish larvae [7,55,56,57]. These apoptotic effects are often accompanied by disrupted neuronal function and altered gene expression related to neurodevelopmental processes. In our previous investigation [9], the results demonstrate that TCDD adversely affects nitrergic neurons and thereby contributes to delayed intestinal motility. In our current study, while studying the mechanism underneath this neuronal loss, we noticed that TCDD affects cell survival by inducing apoptosis in enteric neurons. The observed increase in cleaved caspase-3 and cleaved PARP levels in IM-FEN cells treated with TCDD is AHR-mediated and is consistent with the apoptotic effects reported in various cell types exposed to TCDD, thereby providing valuable insights into its toxicological mechanisms [58,59,60,61]. Furthermore, the use of the AHR antagonist CH-223191 and cells from neural crest-specific *Ahr^−/−^* mice (*Wnt1Cre^+/−^/Ahr^b(fl/fl^*^)^) strengthens the evidence for AHR-mediated mechanisms in TCDD-induced cytotoxicity and apoptosis.

Studies have shown that ER stress can lead to apoptosis with subsequent dysregulation of pro-survival and pro-apoptotic gene expression and the possible role of ER stress in neurodegenerative diseases [24,62,63,64]. Duan et al., studied the role of ER stress in 200 nM TCDD-induced apoptosis in pheochromocytoma (PC12) cells and primary neurons [65]. They found that TCDD-induced ER stress activated the PERK-eIF2α signaling pathway and triggered apoptosis in PC12 cells. However, our findings suggest that TCDD-induced apoptosis in IM-FEN cells occurs through an ER stress-independent mechanism, as the analysis of ER stress markers did not show significant changes in TCDD-exposed IM-FEN cells. This distinction could stem from differences in cell type, TCDD concentrations, or both, underscoring the complexity of the cytotoxic effects of TCDD.

After excluding ER stress as the triggering mechanism for apoptosis in TCDD-exposed IM-FEN cells, we focused on investigating the potential involvement of ceramides. Ceramides have been recognized as important regulators of apoptosis [66,67,68]. They can form large protein-permeable channels in the outer membranes of mitochondria, allowing the release of proapoptotic proteins during the initiation of apoptosis [69]. Ceramide accumulation has been observed, leading to caspase activation during apoptosis [70,71] and transcription of apoptosis-related gene products such as Fas ligand and TNFα [72,73]. Recent studies have shown that POPs like TCDD, PCB, and TCDF can regulate sphingolipid metabolism [46,47]. However, the specific role of AHR in ceramide-induced apoptosis is not well understood.

The enzyme serine palmitoyltransferase (SPT) catalyzes the first, rate-limiting step in the de novo ceramide synthesis pathway and is encoded by the genes *Sptlc1* and *Sptlc2*. SPT is likely active as a heterodimer formed with SPT1 and SPT2 [74], with SPT1 primarily playing a role in stabilizing SPT2 rather than contributing to catalytic activity. This suggests that *Sptlc2* (rather than *Sptlc1*) expression is positively associated with ceramide production. Ceramide is also produced by the hydrolysis of sphingomyelin catalyzed by N-SMase (*Smpd2*, *Smpd3,* and *Smpd4*) in the cell membranes. In our study, chromatin immunoprecipitation (ChIP) assays identified *Sptlc2* and *Smpd2* as potential gene targets of AHR involved in ceramide synthesis. Notably, our results revealed a significant increase in the expression of AHR target genes associated with the de novo ceramide synthesis pathway (*Sptlc1*, *Sptlc2*, *Cers2*, Cers5, *Cers6*, and *Degs1*) and the sphingomyelinase pathway (*Smpd2*, *Smpd3,* and *Smpd4*). These changes in gene expression correlated with alterations in the enzymatic activity of N-SMase, resulting in significant changes in ceramide levels and accumulation. This finding expands our understanding of the biological roles of AHR beyond its involvement in detoxification and xenobiotic metabolism, suggesting a broader regulatory role in cellular lipid homeostasis.

Although we did not directly investigate mitochondrial function in terms of ceramide channels or examine the levels of proapoptotic proteins, we evaluated the impact of ceramides on cell survival and apoptosis in IM-FEN cells using exogenous cell-permeable ceramide analogs, C_2_- and C_6_-ceramides. Short-chain ceramide analogs (C_2_- and C_6_- ceramides) have been used to mimic ceramide-mediated stress-induced cell death pathways in previous studies [69,75]. Physiological concentrations of ceramides can vary significantly depending on the tissue type, metabolic state, and presence of pathological conditions. The dose selection for C_2_- and C_6_-ceramides (25 µM) in the study was determined through a comprehensive review of existing literature that investigated the impact of ceramides on neuronal apoptosis [29,76]. These concentrations have been shown to have significant biological effects in previous studies. The differential time points of cytotoxicity and apoptosis effects observed in the study could be attributed to the underlying mechanisms of the assays used to measure these parameters with varying kinetics. LDH is a cytoplasmic enzyme present in viable cells, and its release indicates compromised cell membrane integrity and cellular damage. The time points at which the observed significant LDH release occurs (within the first 1–6 h) suggest that the short-chain ceramides (C_2_-ceramide and C6-ceramide) rapidly induce damage to the cell membrane, leading to the leakage of LDH. Cytotoxicity may primarily result from acute damage to cellular structures. On the other hand, the differences in the timing of caspase 3/7 activation observed with C_2_-ceramide and C_6_-ceramide treatments (3–6 h and 6–12 h, respectively) suggest that they induce apoptosis through distinct mechanisms or pathways, which may require additional intracellular processes, such as ceramide metabolism or signaling cascades, and that could explain the delayed onset of apoptosis compared to cytotoxicity. The variations in the time points of cytotoxicity and apoptosis could be attributed to subtype-specific effects of these ceramides on the cellular processes involved. Our results indicate that exposure to short-chain ceramides, C_2_-ceramide, and C_6_-ceramide, induces cytotoxicity and increased caspase 3/7 activity, highlighting the induction of apoptosis in IM-FEN cells. Furthermore, our findings demonstrated that co-treatment of TCDD with myriocin, an inhibitor of ceramide synthesis, restored cell survival, further supporting the role of ceramides in apoptosis. It is important to note that while the chosen concentration of ceramides provides valuable insights into ceramide function, it may not fully replicate the physiological environment. Future studies employing a range of ceramide concentrations would be beneficial to better understand the dose-response relationship and to correlate in vitro findings more accurately with in vivo conditions.

Among the recognized bioactive signaling molecules, sphingolipid metabolites play important roles in signal transduction and cell regulation [77,78]. Ceramides can modulate cell death by inhibiting the PI3K antiapoptotic/prosurvival pathway, which involves the regulation of AKT activity [44]. However, the specific role of AHR in ceramide-induced apoptosis is not well understood. Phosphatidylinositol 3-kinase (PI3K) and its downstream effector, the protein-serine/threonine kinase AKT, a negative regulator of GSK-3β, play an important role in preventing apoptosis by blocking the activation of the caspase cascade in several cell types, including neuronal and glial cells [29,79,80,81,82,83]. Reports have suggested that ceramides affect the PI3K/AKT pathway [52,53,72,84]. In our research, we aimed to investigate the underlying molecular mechanism responsible for neuronal cell death induced by TCDD and ceramide, with a specific focus on the PI3K/AKT/GSK-3β pathway. Our findings reveal a significant reduction in the phosphorylation levels of both AKT and GSK-3β in TCDD-treated and C_2_ ceramide-treated IM-FEN cells. This implies that TCDD and ceramides adversely impact the activation and functioning of AKT and GSK-3β in enteric neuronal cells, thereby shedding light on the potential mechanisms contributing to the observed neuronal loss associated with TCDD exposure.

In summary, this study addresses a critical gap in our understanding of TCDD-induced enteric neurotoxicity by elucidating the underlying molecular mechanisms and identifying potential targets for therapeutic intervention. Our findings lay the groundwork for future studies exploring the regulatory role of AHR in apoptosis and sphingolipid metabolism. This could lead to a broader understanding of the environmental factors influencing neurodegenerative diseases and other conditions associated with dysregulated apoptosis. The primary limitation of our study is its focus on in vitro models (IM-FEN cells). While these models provide valuable insights into cellular mechanisms, the findings need validation in in vivo models to establish their physiological relevance and the potential for therapeutic applications. Our study demonstrates a novel pathway for TCDD-induced apoptosis via ceremide accumulation (Figure 8). Future studies will examine additional alternative pathways that could also contribute to TCDD-induced apoptosis. Further research could provide a more comprehensive understanding of the cellular response to TCDD exposure. While the study identifies Sptlc2 and Smpd2 as AHR targets in the regulation of ceramide synthesis, the potential existence of other critical targets or mechanisms involved in this process is not fully explored. Future studies could uncover additional layers of regulation. The study suggests potential therapeutic targets for mitigating TCDD-induced toxicity, but it does not provide direct evidence of efficacy in preventing or reversing the effects of TCDD exposure. Further research is needed to develop and test specific interventions based on these targets. Despite these limitations, our study makes a significant contribution to our understanding of the mechanisms by which environmental pollutants induce neurotoxicity and opens new avenues for research and therapeutic development.

## 4. Materials and Methods

### 4.1. Mice

*Ahr* floxed mice (*Ahr^(fl/fl)^*) were generated by Cyagen Biosciences, Inc., by flanking exons 5 to 7 with *lox*P sites. *Ahr^(fl/fl)^* mice were subsequently backcrossed 10 generations to C57Bl/6J mice and confirmed by the Genome Scanning Services of the Jackson Laboratory to have >99% C57Bl/6J background (*Ahr^b(fl/fl)^*). *Wnt1-Cre* transgenic mice that express Cre recombinase (*B6.Cg-E2f1^Tg(Wnt1−cre)2Sor^/J*) were purchased from Jackson Laboratory. *Ahr* was specifically deleted in neural crest cells by crossing *Ahr^b(fl/fl)^* mice with *Wnt1-cre* mice. The offspring were genotyped using primers (Integrated DNA Technologies, Coralville, IA, USA) listed in Table 3 to confirm that they were homozygous for the *Ahr^b^* floxed gene and either hemizygous (*Wnt1Cre^+/−^/Ahr^b(fl/fl)^*) or negative for *Wnt1-cre* (*Wnt1Cre^−/−^/Ahr^b(fl/fl)^*) mice. *Wnt1Cre^−/−^/Ahr^b(fl/fl)^* mice were used as controls. All experimental mice used were 8- to 10-week-old male mice. Animal studies were approved by the Institutional Animal Care and Use Committee at Pennsylvania State University and performed according to the university guidelines for the ethical treatment of animals.

### 4.2. Cell Culture

In vitro experiments were conducted using Immortal fetal enteric neuronal cells [48]. These cells were cultured in a modified N2 medium [85] prepared with DMEM/F-12 (Thermo Fisher Scientific, Waltham, MA, USA) containing 10 ng/mL glial cell line-derived neurotrophic factor (GDNF) (Shenandoah Biotechnology, Inc., Warminster, PA, USA), 10% fetal bovine serum (FBS) (Bio-Techne, Minneapolis, MN, USA), and 20 U/mL recombinant mouse interferon-γ (MilliporeSigma, Burlington, MA, USA). The culture was maintained in a humidified tissue culture incubator with 10% CO2 at a permissive temperature of 33 °C for 24 to 48 h, allowing the cells to proliferate until confluent monolayers were formed. Subsequently, the medium was replaced with neurobasal-A medium (NBM) supplemented with B-27 serum-free supplement (Thermo Fisher Scientific, Waltham, MA, USA), 1 mmol/L glutamine (Thermo Fisher Scientific, Waltham, MA, USA), 1% FBS, and 10 ng/mL GDNF. The cells were then transferred to an atmosphere of 5% CO2 at 39 °C to allow differentiation for 48 h. After differentiation, the cells were cultured on poly-D-lysine and laminin-coated plates (Corning, NY, USA) and treated with various doses of TCDD (0.1, 1, and 10 nM) (Cambridge Isotope Laboratories, Inc., Tewksbury, MA, USA) at different time intervals (3, 6, 12, and 24 h).

### 4.3. Primary Culture Preparation

Primary cultures of mouse ileum and colonic myenteric neurons were prepared following previously published protocols [86,87], using 8-week-old *Wnt1Cre^+/−^/Ahr^b(fl/fl)^* and *Wnt1Cre^−/−^/Ahr^b(fl/fl)^* mice. Myenteric neurons were seeded onto poly-L-lysine-coated 12 mm coverslips and cultured at 37 °C with 5% CO_2_ in complete NBM, as described earlier. The medium was partially replaced every 24 h. After 5 days in culture, the neurons were treated with 10 nM TCDD for 24 h.

### 4.4. LDH-Glo Cytotoxicity Assay

Immorto fetal enteric neuronal cells were cultured on 96-well white plates with clear bottoms (10,000 cells/well) at 39 °C and treated with vehicle, 10 nM TCDD with or without 10 µM CH-223191 (MilliporeSigma, Burlington, MA, USA), a potent and specific antagonist of AHR, and 10 µM Myriocin (MilliporeSigma, Burlington, MA, USA), a potent inhibitor of SPT, the rate-limiting enzyme of ceramide biosynthesis, and 25 µM C_2_- and C_6_-ceramides and dihydroceramides (Enzo Life Sciences, Inc., Farmingdale, NY, USA) in complete NBM for different time points (0.5, 1, 3, 6, 12, 24 h). Cells were pretreated with 10 µM CH-223191 and 10 µM Myriocin for 1 h prior to the addition of 10 nM TCDD. The percentage of cytotoxicity was calculated relative to the maximum lactate dehydrogenase (LDH) release control (10% Triton^®^ X-100) as recommended by the manufacturer’s instructions using the LDH-Glo Cytotoxicity Assay kit from Promega (Madison, WI, USA).

### 4.5. Caspase-Glo 3/7 Assay

Immorto fetal enteric neuronal cells were cultured on 96-well white plates with clear bottoms (10,000 cells/well) at 39 °C and treated with vehicle, 10 nM TCDD with or without 10 µM CH-223191, 25 µM C_2_- and C_6_-ceramides and dihydroceramides (Enzo Life Sciences, Inc., Farmingdale, NY, USA) in complete NBM for different time points (0.5, 1, 3, 6, 12, 24 h). Cells were pretreated with 10 µM CH-223191 for 1 h prior to the addition of 10 nM TCDD. Caspase 3/7 activity was assessed using the Caspase-Glo 3/7 Assay kit from Promega (Madison, WI, USA), in accordance with the manufacturer’s protocol. Luminometer readings were taken 1 h after the addition of the Caspase-Glo 3/7 Reagent.

### 4.6. Real-Time PCR

Immorto fetal enteric neuronal cells cultured on poly-D-lysine and laminin-coated 6-well plates (160,000 cells/well) at 39 °C were treated with vehicle and 10 nM TCDD at various time points (0.5, 1, 3, 6, 12, 24 h). Total RNA was extracted using TRIzol Reagent (Thermo Fisher Scientific, Waltham, MA, USA) according to the manufacturer’s instructions. Complementary DNA (cDNA) was synthesized from 1 µg of total RNA using the qScript cDNA SuperMix (Quantabio, Beverly, MA, USA). Quantitative real-time PCR was performed with cDNA using PowerUp SYBR Green Master Mix (Applied Biosystems, Waltham, MA, USA) on the QuantStudio 3 Real-Time PCR System (Applied Biosystems, Waltham, MA, USA) with primers listed in Table 1. The Ct values obtained were normalized to the housekeeping gene B2M.

### 4.7. ChIP-PCR/qPCR Assay

The ChIP-PCR/qPCR assay was conducted by the SimpleChIP^®^ Plus Enzymatic Chromatin IP Kit (Magnetic Beads) according to the manufacturer’s instructions (Cell Signaling Technology, Danvers, MA, USA). Briefly, IM-FEN cells cultured on 15 cm dishes (4 × 10^6^ cells/dish) at 39 °C were treated with vehicle and 10 nM TCDD for 1 h were collected, and, proteins and DNA were cross-linked with 1% formaldehyde in PBS at room temperature for 10 min. Chromatin was extracted and immunoprecipitated with 2 μg of antibodies to AHR (Enzo, BML-SA210-0100) at 4 °C overnight. The immune complexes were collected by incubation with protein G magnetic beads, and the beads were washed with low salt, high salt, and ChIP buffer. After the elution of bound chromatin, the genomic DNA was purified using a spin column and subjected to PCR or qPCR using primers specific for the *Cyp1a1*, *Sptlc2*, and *Smpd2* promoter regions, which are known to contain DREs, or the non-binding region as a negative control (Table 2).

### 4.8. Sphingomyelinase Assay

Immorto fetal enteric neuronal cells were cultured in 96-well plates (10,000 cells/well) at 39 °C and treated with vehicle and 10 nM TCDD for 24 h. Neutral sphingomyelinase (N-SMase) activity was measured using a colorimetric sphingomyelinase assay kit from Sigma (Burlington, MA, USA) according to the manufacturer’s protocol.

### 4.9. Orbitrap-MS Analysis of Lipids

IM-FEN cells cultured on 10 cm dishes (2 × 10^6^ cells/dish) at 39 °C were treated with vehicle, and 10 nM TCDD for 3 h were collected and extracted twice with 1.0 mL of pre-cooled isopropanol/water/ethyl acetate (30:10:60, *v*:*v*:*v*) containing 1:1000 Equisplash (Avanti Polar Lipids, Alabaster, AL, USA). After sonication and centrifugation (Eppendorf, Hamburg, Germany), the supernatant was collected, evaporated to dryness (SpeedVac, Thermo Scientific, Waltham, MA, USA), and dissolved in 60 μL of isopropanol/acetonitrile/H_2_O (45:35:20, *v*:*v*:*v*). After centrifugation, the supernatants were transferred to autosampler vials for LC-MS analysis.

Total lipids were analyzed by a Vanquish UHPLC system coupled to an Orbitrap Fusion Lumos Tribrid™ mass spectrometer using a H-ESI™ ion source (all Thermo Fisher Scientific) with a Waters (Milford, MA, USA) CSH C18 column (1.0 × 150 mm; 1.7 µm particle size) as previously described [88]. LC-MS data were analyzed by the open-source software MS-DIAL ver. 4.9.221218 [89], identifications were performed using the in-silico lipid library, and the spectra were manually checked to confirm assignments. The experiment was processed in technical duplicate, and the averages were used for statistical purposes.

### 4.10. Western Blotting

Cell lysates were obtained from IM-FEN cells cultured on poly-D-lysine and laminin-coated 6-well plates (160,000 cells/well) at 39 °C and treated with or without TCDD (0.1, 1, and 10 nM) and C_2_ ceramide (25 µM) for 24 h. Additionally, some cells were pre-treated with the AHR antagonist CH-223191 (10 µM) for one hour before a 24-h exposure to 10 nM TCDD. Thapsigargin (300 nM, 5-h treatment) (MilliporeSigma, Burlington, MA, USA) was used as a positive control to benchmark ER stress. Proteins (15 µg per lane) were loaded onto a 4–20% sodium dodecyl sulfate–polyacrylamide gel electrophoresis (Bio-Rad, Hercules, CA, USA) and transferred to a PVDF membrane. The membranes were probed for Cleaved caspase-3, Cleaved PARP, Phospho-Akt (Ser473), Akt, Phospho-GSK-3β (Ser9), GSK-3β, and ER stress-associated proteins, including GRP78, IRE1α, and CHOP, using respective specific antibodies (Table 4). GAPDH was used as a loading control. Band intensity was measured semiquantitatively using the Scion Image software (https://scion-image.software.informer.com/ accessed on 25 July 2024) (Scion Corporation, Frederick, MD, USA) and expressed as a ratio of band intensity relative to the loading control.

### 4.11. TUNEL Staining

Immorto fetal enteric neuronal cells cultured on poly-L-lysine coated 12 mm coverslips (20,000 cells/well) (Neuvitro Corporation, Camas, WA, USA) were treated with vehicle and 10 nM TCDD for 24 h. To detect apoptosis, terminal deoxynucleotidyl transferase dUTP nick end labeling (TUNEL) staining was performed using the In Situ Cell Death Detection Kit, Fluorescein (Roche Diagnostics, Indianapolis, IN, USA), following the manufacturer’s protocol. The cells were counterstained with enteric neuronal marker TUJ1 in conjunction with Alexa Fluor 594 (Table 4) and nuclear dye DAPI (AnaSpec, Fremont, CA, USA). Images were captured on a Revolve R4 Microscope (ECHO, San Diego, CA, USA).

### 4.12. Immunofluorescence Staining of Primary Culture Cells

Myenteric neurons from *Wnt1Cre^+/−^/Ahr^b(fl/fl)^* and *Wnt1Cre^−/−^/Ahr^b(fl/fl)^* mice were cultured on poly-L-lysine coated 12 mm coverslips (20,000 cells/well) and treated with vehicle or 10 nM TCDD for 24 h at 37 °C. Following treatment, the cells were fixed using 4% paraformaldehyde for 20 min and then blocked for 1 h with a blocking and permeabilization solution containing 0.3% Triton X-100 and 5% BSA (MilliporeSigma, Burlington, MA, USA) in PBS. The cells were then incubated with primary antibodies (TUJ1 and cleaved caspase-3, as listed in Table 4) at room temperature for 2 h. Secondary detection was performed in conjunction with Alexa Fluor 594 or 488. Images were captured with a Revolve R4 Microscope (ECHO, CA).

### 4.13. Statistical Analysis

Data analysis was performed using GraphPad Prism version 9.1.1 (GraphPad, San Diego, CA, USA). Statistical analysis for two treatment groups was conducted using the Student’s *t*-test. For multiple data sets, one-way analysis of variance (ANOVA) was employed, followed by Tukey’s multiple comparison test. Results are presented as mean ± SEM, with a *p*-value of 0.05 or less considered statistically significant.

## Figures and Tables

**Figure 1 ijms-25-08581-f001:**
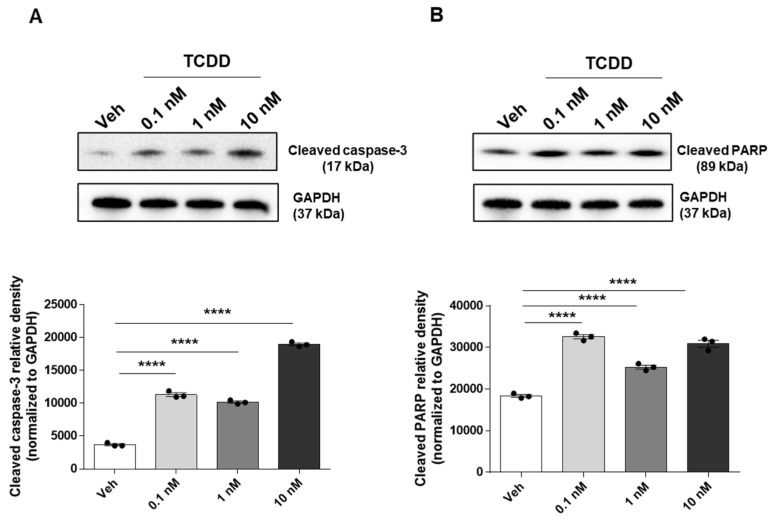
TCDD induces apoptosis in IM-FEN cells. Cells were treated with vehicle or various doses of TCDD (0.1, 1, and 10 nM) for 24 h. Representative images of cleaved caspase-3 and cleaved PARP Western blots are included (**A**,**B**). Results from the protein bands’ density normalized to GAPDH have been included in the correspondent graphs. (**C**) Representative images of neuronal marker TUJ1 (red), TUNEL (green), and DAPI (blue) immunostaining in IM-FEN cells treated with vehicle and 10 nM TCDD for 24 h. Arrows point to the apoptotic cells (yellow) with condensed nuclei. Magnified image of the neuron shows the DNA fragmentation seen during apoptosis. Scale bar, 90 μm. The data represent three independent experiments. Results are mean ± SEM, **** *p* < 0.0001.

**Figure 2 ijms-25-08581-f002:**
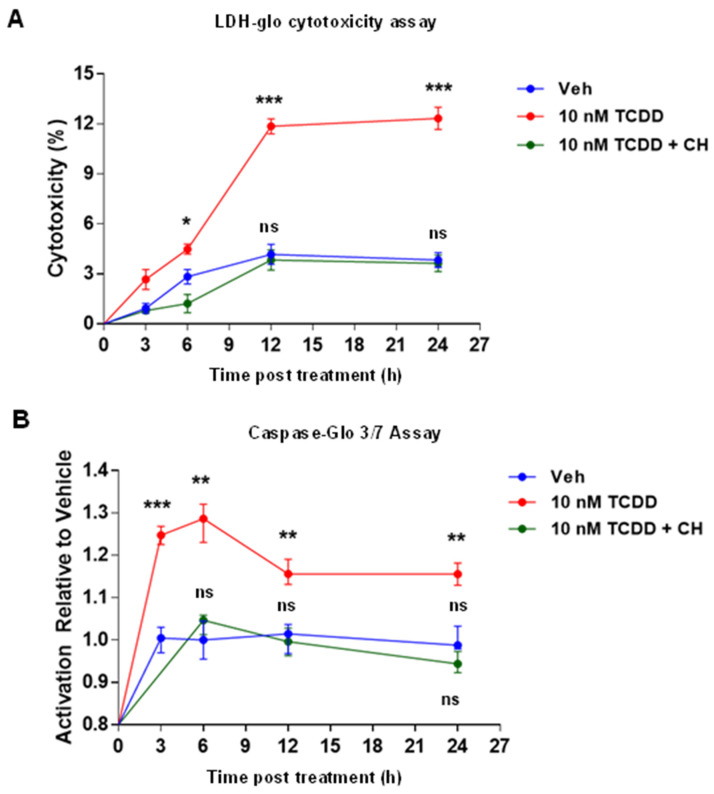
Cytotoxicity and apoptosis induced by TCDD are AHR-dependent. IM-FEN cells were pretreated with an AHR antagonist, CH-223191 (10 µM), for 1 h and then treated with 10 nM TCDD at different time points (3, 6, 12, and 24 h). (**A**) Cytotoxicity was assessed by an LDH release assay. The percentage of cytotoxicity was calculated relative to the maximum LDH release control (10% Triton^®^ X-100). (**B**) Cell death by apoptosis was assessed by measuring Caspase-3/7 activity 1 h after adding the Caspase−Glo-3/7 reagent. Statistical analysis of LDH cytotoxicity assay data shows significant differences between vehicle and 10 nM TCDD after 6, 12, and 24 h of treatment. Not significant (ns) differences were observed between the vehicle and 10 nM TCDD + CH-223191 experimental groups. Results from the Caspase 3/7 assay show significant differences between vehicle and 10 nM TCDD treatment at all timepoints studied, whereas n.s. differences were observed when comparing the vehicle and 10 nM TCDD + CH-223191 experimental groups. Apoptosis was assessed by TUJ1/cleaved caspase-3 immunostaining of myenteric neurons isolated from (**C**) *Wnt1Cre^−/−^/Ahr^b(fl/fl)^* (control) mice and (**D**) *Wnt1Cre^+/−^/Ahr^b(fl/fl)^* (neural crest-specific *Ahr^−/−^*) mice treated with vehicle and 10 nM TCDD for 24 h. Representative images show TUJ1 (red) and cleaved caspase-3 (green). The white arrow points to Tuj1^+^ cell bodies, and the white arrowhead points to the axons. The yellow arrowhead points to cleaved caspase-3-positive neurons. Scale bar, 90 μm. Results are mean ± SEM; * *p* < 0.05; ** *p* < 0.01; *** *p* < 0.001.

**Figure 3 ijms-25-08581-f003:**
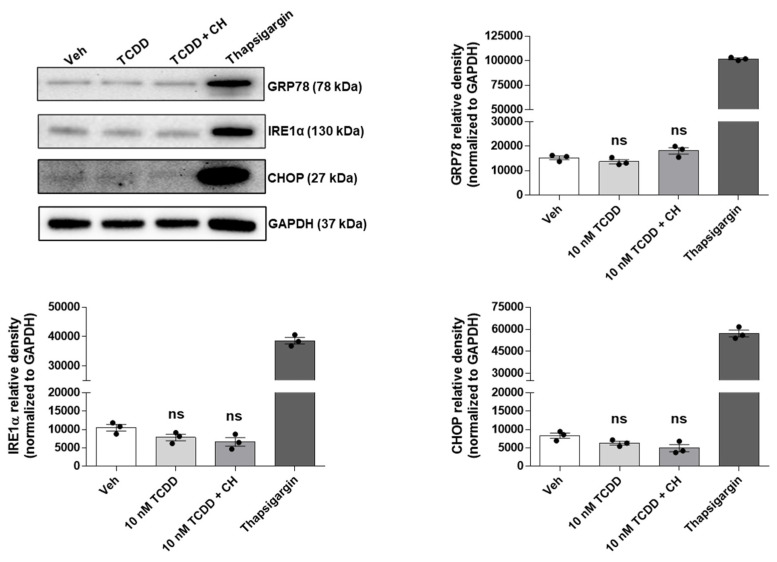
TCDD-induced apoptosis in IM-FEN cells is independent of ER stress. Cells were treated with Veh and 10 nM TCDD for 24 h. Some cells received pre-treatment with the AHR antagonist CH-223191 (10 µM) for one hour before a 24 h exposure to 10 nM TCDD. Thapsigargin (300 nM, 5 h treatment) was used as a positive control for ER stress. Representative images of GRP78, IRE1α, and CHOP Western blots are included. Results from the protein bands’ density normalized to GAPDH have been included in the correspondent graphs. Results are mean ± SEM. Not significant (ns) differences were observed between vehicle and 10 nM TCDD with or without CH-223191 experimental groups.

**Figure 4 ijms-25-08581-f004:**
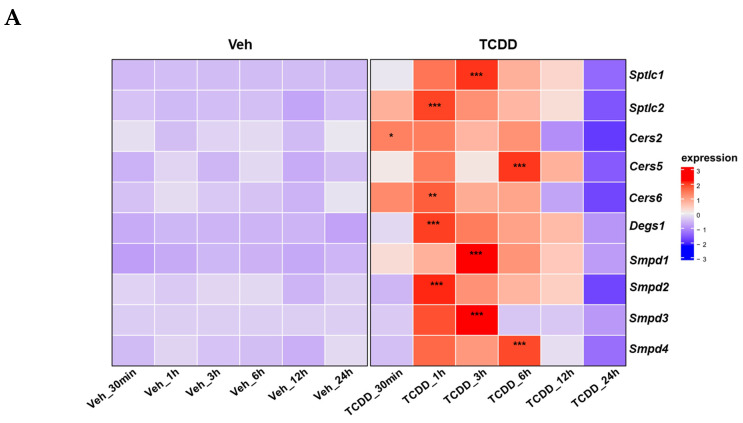
Regulation of ceramide biosynthesis and sphingolipid metabolism by TCDD in IM-FEN cells. Quantitative PCR (qPCR), neutral sphingomyelinase (N-SMase) activity assays, lipidomics, and chromatin immunoprecipitation (ChIP) assays to elucidate the effects of TCDD treatment on ceramide metabolism and gene regulation. (**A**) Heat map visualization shows the normalized expression levels of key genes involved in ceramide biosynthesis (*Sptlc1*, *Sptlc2*, *Cers2*, *Cser5*, *Cers6*, *Degs1*, *Smpd1*, *Smpd2*, *Smpd3*, and *Smpd4*) in IM-FEN cells treated with vehicle or 10 nM TCDD over various time points (30 min, 1, 3, 6, 12, and 24 h), with upregulated genes in red and downregulated genes in blue. Columns represent the timepoint, and rows represent individual genes. (**B**) N-SMase activity of vehicle and 10 nM TCDD-treated IM-FEN cells for 24 h. (**C**) Lipidomics analysis presented through a heatmap, illustrating the log2 mean-centered normalized data of sphingolipids in vehicle and 10 nM TCDD-treated IM-FEN cells, with tiles colored red for high abundance and blue for low abundance. (**D**) Statistically significant changes in sphingolipids, sorted into categories, are highlighted to show the alterations in ceramide, sphingomyelin, and hexosylceramide levels. log2 mean-centered data were imported into R, and the Complex Heatmap package was used to create the heatmap. (**E**,**F**) IM-FEN cells were treated with vehicle and 10 nM TCDD for 1 h. (**E**) ChIP-qPCR result for *Cyp1a1* gene promoter. (**F**) ChIP-qPCR results for *Sptlc2* and *Smpd2* gene promoters that were quantified by normalization with the corresponding input signal. Results are mean ± SEM, * *p* < 0.05; ** *p* < 0.01; *** *p* < 0.001.

**Figure 5 ijms-25-08581-f005:**
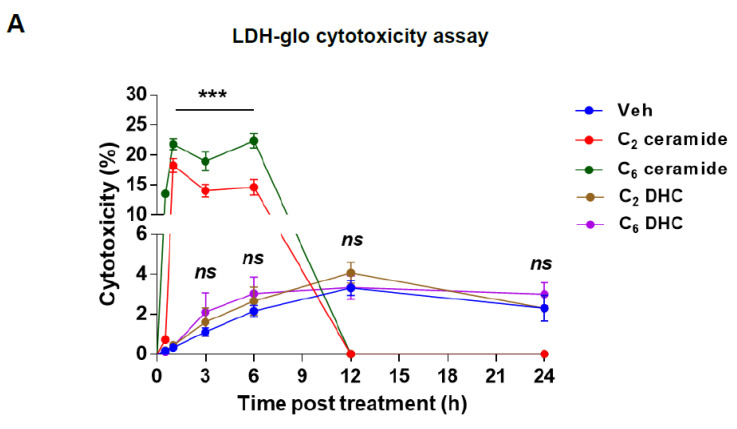
Short-Chain Ceramides induce cytotoxicity and apoptosis in IM-FEN cells. IM-FEN cells were treated with vehicle, 25 µM C_2_-ceramide, C_6_-ceramide, C_2_-DHC, and C_6_-DHC for 30 min, 1-, 3-, 6-, 12-, and 24-h. (**A**) Cytotoxicity was assessed by the LDH release assay. The percentage of cytotoxicity was calculated relative to the maximum LDH release control (10% Triton^®^ X-100). (**B**) Apoptosis was assessed by measuring Caspase 3/7 activity. The statistical significance of the C_2_-ceramide and C_6_-ceramide treatment groups is shown with respect to vehicle. Not significant (ns) differences were observed between the vehicle and C_2_- and C_6_-DHC experimental groups. Results are mean ± SEM, * *p* < 0.05; ** *p* < 0.01; *** *p* < 0.001.

**Figure 6 ijms-25-08581-f006:**
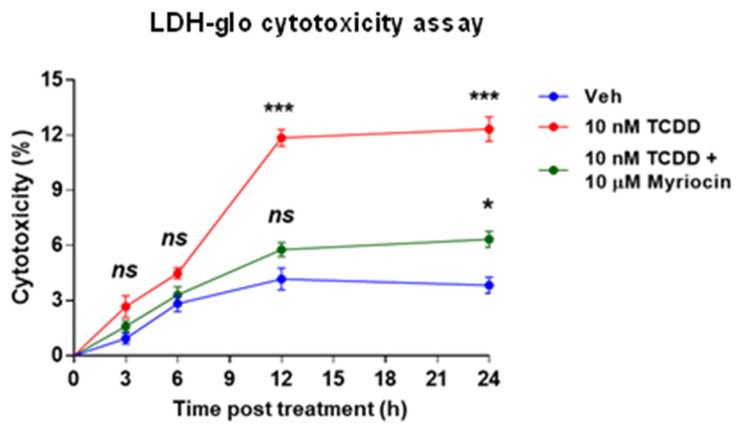
Ceramide involvement in TCDD-induced cytotoxicity in IM-FEN cells. Cells were pretreated with 10 µM myriocin, an inhibitor of ceramide synthesis, for an hour and then treated with 10 nM TCDD at different time points (3, 6, 12, and 24 h). Cytotoxicity was assessed by the LDH release assay. The percentage of cytotoxicity was calculated relative to the maximum LDH release control (10% Triton^®^ X-100). The statistical significance of 10 nM TCDD with or without myriocin is shown with respect to vehicle. Results are mean ± SEM, * *p* < 0.05; *** *p* < 0.001. ns: Not significant.

**Figure 7 ijms-25-08581-f007:**
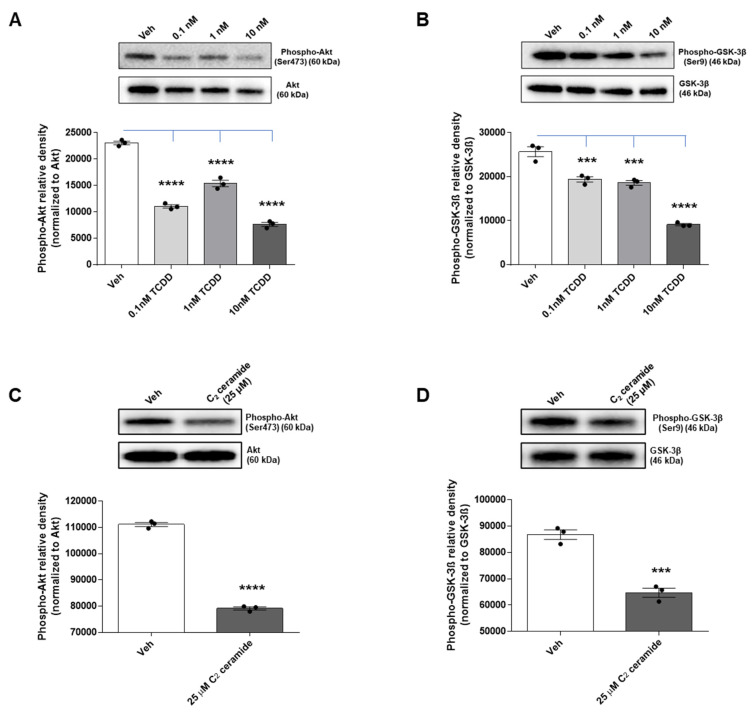
Impact of TCDD and ceramide on neuronal survival pathways via Akt and GSK-3β modulation. IM-FEN cells were treated with vehicle and various doses of TCDD (0.1, 1, and 10 nM) and C_2_ ceramide (25 µM) for 24 h. Representative images of Phospho-Akt (**A**,**C**) and Phospho-GSK-3β (**B**,**D**) Western blots are included. Results from the protein bands’ density normalized to corresponding Akt and GSK-3β have been included in the correspondent graphs. Results are mean ± SEM, *** *p* < 0.001; **** *p* < 0.0001.

**Figure 8 ijms-25-08581-f008:**
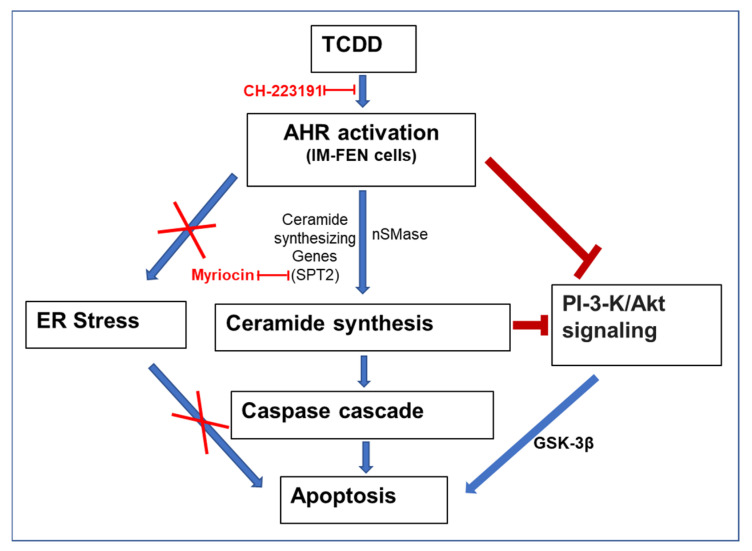
A proposed model showing enhanced ceramide synthesis leading to apoptosis in IM-FEN cells. Key components showing activators in blue arrows and inhibitory associations in red lines.

**Table 1 ijms-25-08581-t001:** Primers used for Real-time PCR.

Gene	Forward Primer (5′→3′)	Reverse Primer (5′→3′)
*B2M*	CATGGCTCGCTCGGTGAC	CAGTTCAGTATGTTCGGCTTCC
*Sptlc1*	CGAGGGTTCTATGGCACATT	GGTGGAGAAGCCATACGAGT
*Sptlc2*	TCACCTCCATGAAGTGCATC	CAGGCGTCTCCTGAAATACC
*Cers2*	AAGTGGGAAACGGAGTAGCG	ACAGGCAGCCATAGTCGTTC
*Cers5*	CTTCTCCGTGAGGATGCTGT	GTGTCATTGGGTTCCACCTT
*Cers6*	AAGCCAATGGACCACAAACT	TGCTTGGAGAGCCCTTCTAAT
*Degs1*	AATGGGTCTACACGGACCAG	TGGTCAGGTTTCATCAAGGAC
*Smpd1*	GTTACCAGCTGATGCCCTTC	AGCAGGATCTGTGGAGTTG
*Smpd2*	GACATCCCCTACCTGAGCAA	CCAGGAGAGCCAGATCAAAG
*Smpd3*	CCTGACCAGTGCCATTCTTT	AGAAACCCGGTCCTCGTACT
*Smpd4*	ACCTGGCCCTCAATCCATTTG	ATAGGCACAGTCCGAAGTACG

**Table 2 ijms-25-08581-t002:** Primer sequences for ChIP-PCR (qPCR).

Gene	Abbreviation	Sequence
Cytochrome P450, family 1, subfamily A, polypeptide 1	*Cyp1a1*	CAGAGGATGGAGCAGGCTTA AGGATCCACGCGAGACAG
Serine palmitoyltransferase, long chain base subunit 2	*Sptlc2*	ACCTCTCCGAAACCGGAAAT GCGAGCCCCGTCTTCTC
Sphingomyelin phosphodiesterase 2	*Smpd2*	AGAGAGGGTTGTGTGTGTGC GATGATGAAGTTGGCAGAGC
Negative control		CAGCAGCTCTTTGTGCTGAC TCCATCTGTGCAGCCTGTAG

**Table 3 ijms-25-08581-t003:** Primers used for genotyping.

Gene	Forward Primer (5′→3′)	Reverse Primer (5′→3′)
*Ahr*	AAGGCCACTAAAGCAATGGGATGT	AAAGTTTATGACTGGGTCACCACA
*Wnt1Cre*	CAGCGCCGCAACTATAAGAG	CATCGACCGGTAATGCAG

**Table 4 ijms-25-08581-t004:** Antibodies used in Western blot (WB) and immunofluorescence (IF).

Antibody/Host	Company	Cat. No.	Dilution
Cleaved Caspase-3 (Asp175)	Cell Signaling Technology (Danvers, MA, USA)	9661	1:1000 (WB) 1:200 (IF)
Cleaved PARP (Asp214)	Cell Signaling Technology	94885	1:1000 (WB)
Phospho-Akt (Ser473)	Cell Signaling Technology	9271	1:1000 (WB)
Akt	Cell Signaling Technology	9272	1:1000 (WB)
Phospho-GSK-3β (Ser9)	Cell Signaling Technology	9336	1:1000 (WB)
GSK-3β	Cell Signaling Technology	12456	1:1000 (WB)
BiP (C50B12)	Cell Signaling Technology	3177	1:1000 (WB)
IRE1α (14C10)	Cell Signaling Technology	3294	1:1000 (WB)
CHOP (D46F1)	Cell Signaling Technology	5554	1:1000 (WB)
GAPDH/Rabbit	Cell Signaling Technology	2118	1:2500 (WB)
TUJ1/Mouse	abcam (Waltham, MA, USA)	ab78078	1:500 (IF)
HRP-conjugated anti-mouse	Cell Signaling Technology	7076	1:2000 (WB)
HRP-conjugated anti-rabbit	Cell Signaling Technology	7074	1:2000–5000 (WB)
Goat anti-Rb, Alexa Fluor 488	Invitrogen (Waltham, MA, USA)	A11008	1:200 (IF)
Goat anti-Mouse, Alexa Fluor 594	Invitrogen (Waltham, MA, USA)	A11005	1:200 (IF)

## Data Availability

Data is contained within the article.

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
