# Peer review of "Modulation of Ceramide-Induced Apoptosis in Enteric Neurons by Aryl Hydrocarbon Receptor Signaling: Unveiling a New Pathway beyond ER Stress"

_ijms, 2024, doi:10.3390/ijms25168581_

Round 1

Reviewer 1 Report

Comments and Suggestions for Authors

In the present manuscript Mallapa and al. study the ceramide-induced apoptosis in enteric neurons by AHR signaling. To be consider for publication authors should clarify:

Cell Types Used in Experiments: The authors should clarify which cell types they used in their experiments. Although they described deriving primary cultures from two mouse strains and an IM-FEN cell line, it appears that they consistently used the immortalized cell line. This discrepancy should be clarified.

Correlation of TCDD Doses with Practical Exposure in Humans: The authors need to address how the tested doses of TCDD correlate with practical exposure doses in humans. TCDD is a persistent organic pollutant and understanding the relevance of the doses used in their study to real-world exposure is crucial.

Figure 5: Cytotoxicity and Caspase Activity: In Figure 5, the % of cytotoxicity and caspase activity for treatment with C2 and C6 ceramide reaches zero. The explanation for this phenomenon should be provided. Is it because all cells are already dead, or is there another reason? Further details are needed to interpret these results accurately.

Ceramide Concentration in Treatment: The authors should specify the concentration of ceramides used in their treatment experiments. Understanding the dosage is essential for assessing the impact on cellular processes. Lastly, the authors should discuss how the ceramide concentrations used in their study correlate with the physiological state in the body. Does the observed effect align with what might occur under normal conditions?

Reviewer 2 Report

Comments and Suggestions for Authors

Coments have been included in the attached pdf file

Round 2

Reviewer 2 Report

Comments and Suggestions for Authors

Authors have considered all reviwers' sugestions thus

much improving the manuscript.

 In this reviwer's opinion the manuscript deserves to be published.